# Synthesis of Lightweight Renewable Microwave-Absorbing Bio-Polyurethane/Fe_3_O_4_ Composite Foam: Structure Analysis and Absorption Mechanism

**DOI:** 10.3390/ijms232012301

**Published:** 2022-10-14

**Authors:** Xiaoling Xu, Xiaoke Tian, Guangxu Bo, Xingjian Su, Jinyong Yan, Yunjun Yan

**Affiliations:** Key Laboratory of Molecular Biophysics of the Ministry of Education, College of Life Science and Technology, Huazhong University of Science and Technology, Wuhan 430074, China

**Keywords:** bio-based polyurethane, lightweight magnetic foam, hierarchical microstructure, microwave absorption mechanism

## Abstract

Sustainable renewable polymer foam used as a lightweight porous skeleton for microwave absorption is a novel strategy that can effectively solve the problems of the large surface density, high additive amount, and narrow absorbing band of absorbing materials. In this article, novel renewable microwave-absorbing foams were prepared using *Sapiumse biferum* kernel oil-based polyurethane foam (BPUF) as porous matrix and Fe_3_O_4_-nanoparticles as magnetic absorbents. The microstructure and the microwave absorption performance, the structural effects on the properties, and electromagnetic mechanism of the magnetic BPUF (mBPUF) were systematically characterized and analyzed. The results show that the mBPUF displayed a porous hierarchical structure and was multi-interfacial, which provided a skeleton and matching layer for the Fe_3_O_4_ nanoparticles. The effective reflection loss (*RL* ≤ −10 dB) frequency of the mBPUF was from 4.16 GHz to 18 GHz with only 9 wt% content of Fe_3_O_4_ nanoparticles at a thickness of 1.5~5 mm. The surface density of the mBPUF coatings was less than 0.5 kg/cm^2^ at a thickness of 1.8 mm. The lightweight characteristics and broadband absorption were attributed to the porous hierarchical structures and the dielectric combined with the magnetic loss effect. It indicates that the mBPUF is a prospective broadband-absorbing material in the field of lightweight stealth materials.

## 1. Introduction

It is well-known that the porous structure design and lightweight components of materials could significantly influence the attenuation ability and absorption capacity of electromicrowaves [1,2,3]. Microwave absorption materials (MAMs) are one of the most important strategic materials for electromagnetic stealth and protection [4,5]. Polyurethane foams (PUFs) are considered to be one of the most promising lightweight porous materials for directional channels and skeletons, providing an ideal template to accommodate a variety of absorbents [6,7,8]. 

Many works on the microwave-absorbing composites consisting of PUFs and absorbent materials have been reported, and there has been significant progress. For example, Li et al. [9] prepared porous thermoplastic polyurethane/graphene (TPU/G) composites using a facile vapor-induced phase separation (VISP) technique. The effective absorption bandwidth (EBA) below −10 dB of the composite containing 3 wt% graphene (TPU/G-3) achieved 4.28 GHz at a thickness of 3.1 mm. After the incorporation of Fe_3_O_4_ on the TPU/G-3, the minimum reflection loss (*RL*_min_) reached −58.96 dB and the matching thickness was 8.0 mm. It indicated that the porous structure construction of TPU/G was beneficial to obtain excellent absorption ability. Zheng et al. [10] fabricated a carbon nanotube @Fe_3_O_4_/polyurethane (CNTs@Fe_3_O_4_/PU) composite foam-based triboelectric nanogenerator by assembling self-foaming. The effective absorption bandwidth (EAB) (*RL* ≤ −10 dB) was 4.37 GHz at a thickness of 2.55 mm under a filler loading of 15 wt%. The tunable microwave-absorbing mechanism was due to the good impedance matching, high dielectric and magnetic loss, and multiple reflections and scatterings. Gao et al. [11] used TPU as matrix to prepare TPU/G flexible composite foam with different G contents and foam ratios. The effect frequency region of the composite foam was 4.7 GHz at a thickness of 1.6 mm and a graphene content and foam ratio of 0.82 vol% and 3.9, respectively. The high microwave-absorbing performance was attributed to the adjustment of the dielectric permittivity and loss and the alteration of thickness. However, the above-mentioned coating-absorbing materials still have some problems, such as unsustainable raw materials, a complex preparation process, small quantity and high prices, and unsatisfactory broadband-absorbing effects. For these reasons, sustainable lightweight bio-based absorbing composites based on microstructure design and effective compounding are the dominant strategies to obtain high performance and extend the practical applications [12]. Bio-based polyurethane (BPU) and its foams (BPUFs) have been evaluated as polymers that contain lightweight, flexible, corrosion-resistant, easy-molding, and exceptional thermal–mechanical performance in previous research. Therefore, they are competitive substrates compared to traditional petroleum-based PUFs [13,14,15,16]. To date, few works have been concerned with the BPU as a matrix skeleton for the application of microwave absorption. 

Therefore, in this article, a novel BPUF synthesized with *Sapiumse biferum* kernel oil polyol is used as the porous scaffold for embedding magnetic Fe_3_O_4_ nanoparticles. Then, the microstructures, surface and interface characteristics, magnetic properties, and absorption mechanism of the magnetic foam (mBPUF) are characterized and analyzed. Furthermore, the microstructural effects on the properties of the mBPUF are intensively evaluated combined with molecular dynamic simulation. 

## 2. Results

### 2.1. Structure Characterization and Analysis

Figure 1 shows the transmission spectrum of the magnetic Fe_3_O_4_ particle and the mBPUF composites. In the spectrum for the Fe_3_O_4_ particle, the peaks at 3404 cm^−1^, 1635 cm^−1^, and 588 cm^−1^ correlate to the characterization absorption of the O–H and Fe–O groups, respectively [17]. For the mBPUF, the characteristic absorption peaks of the Fe–O groups transfer to 1599 cm^−1^ and 507 cm^−1^, which display an apparent change in the transmittance. This may be due to the molecular interaction of the polar functional groups between Fe_3_O_4_ and the BPUF matrix. The peak at 3368 cm^−1^ is the antisymmetric stretching vibrations of the O–H groups and the N–H groups, which exist in the molecular chain of the BPUF matrix. The typical characteristic absorption peaks seen at 1720 cm^−1^, 1223 cm^−1^, and 1053 cm^−1^ are associated with the urethane C=O, C–N, and C–O stretching modes [18]. 

The XRD patterns of Fe_3_O_4_ and the mBPUF are depicted in Figure 2. The Fe_3_O_4_ nanoparticles in Figure 1 exhibit sharp peaks, which are located at 30.8° (220), 35.5° (311), 36.9° (222), 42° (331), 43.2° (400), 45.9° (331), 53.6° (422), 57.7° (511), and 64° (440) [19,20]. For the mBPUF, the majority of the diffraction peaks for Fe_3_O_4_ are presented, but the intensity of the peaks is weak. This indicates that part of the magnetic absorbents is embedded in the porous resin, which influences the crystalline structure. One reason is that Fe_3_O_4_ particles are masked and cannot be detected in the porous resin matrix. The other reason is that the molecular interaction between the Fe_3_O_4_ particles and BPUF matrix may result in a slight shift in diffraction angle and the difference in the crystalline size. The peak at 22° of the mBPUF is associated with the base polymer. The presence of the macromolecule shell, which wraps the Fe_3_O_4_ particles and changes the chain’s conformational restrictions [21].

The structure of the magnetic Fe_3_O_4_, the fracture structure of the mBPUF, and its foam powders are displayed in Figure 3. Figure 3A shows that the magnetic Fe_3_O_4_structures are octahedral granular nanoparticles [22]. A porous structure for the mBPUF is shown in Figure 3B, which has embedded within it plenty of magnetic particles. These particles are mostly evenly distributed within the foam holes, while some are scattered on the walls or retained within the interior of the pores, as shown in Figure 3D,F. The element features for the structure are exhibited in Figure 3E,G. The tagged spectrum 1 and spectrum 3 are the ones that reveal the characteristics of the magnetic Fe_3_O_4_nanoparticles. Figure 3C is an image of the foam powders, which have an irregular shape. Here, the edges of the powder are smooth after the foam is crushed.

The XPS curves, shown in Figure 4, are used to analyze the surface element characteristics and the chemical compositions of the obtained foam. From the curves, the four elements (Fe, O, N, and C) are observed and the characteristic peaks of the Fe, O, and C elements are analyzed. It can be observed that ferrum with its two oxidation states, Fe 2p_1/2_ and Fe 2p_3/2_, are found at the energies of 724.69 eV and 711.14 eV, which is the characteristic of the Fe_3_O_4_ nanoparticles in the mBPUF [23]. This indicates that the foam powders can be both embedded and encased within the matrix. For the mBPUF, the O 1s signals are constituted by two peaks at 533.05 and 530.24 eV, which relate to a composition of C=O and C–O in the chain of the matrix [24]. The O 1s peaks of the film overlap into almost one peak. The C 1s signals at 288.33, 285.86, 286.59, and 285.06 eV are the characteristics of C=O, C–N, C–O, and C–C/C–H for the mBPUF powder, respectively [25]. The N 1s signal is the characteristic of C–N in the main chain of the BPUF matrix. 

### 2.2. Electromagnetic Parameters and Microwave Absorption Properties 

The magnetic properties of the Fe_3_O_4_and its mBPUF are characterized using a vibrating sample magnetometer (VSM) at room temperature. The magnetization hysteresis loops for the samples are displayed in Figure 5. The saturation magnetization of the Fe_3_O_4_ particles is 98.88 emu/g and the obtained superparamagnetism of the samples contains negligible coercivities. For the mBPUF, the saturation magnetization decreases to 15.18 emu/g and is characterized by the superparamagnetism [26]. 

In the article, the basic transmission line equations are constructed according to transmission line theory and a microwave network. All of these effective parameters can be calculated from the corresponding frequency-dependent parameters for bulk components using electromagnetic field theory. It is the main tool to analyze transmission line problems based on Maxwell’s equations (field method) and circuit theory based on Kirchhoff’s laws (circuit method). The electromagnetic parameters are tested in a vector network analyzer (VNA system), which covers the calculation method of the transmission line theoretical equation. The electromagnetic performance of the mBPUF is a consolidated result of the electromagnetic parameters, which include the complex permittivity (*ε_r_* = *ε*′ − *ε*″) and the complex permeability (*μ_r_* = *μ*′ − j*μ*″). *ε*′, *ε*″ and *μ*′, *μ*″ with frequencyis displayed in Figure 6a. The real part parameters (*ε*′ and *μ*′) denote the storage abilities of electrical energy and magnetic energy, and the imaginary part parameters (*ε*″ and *μ*″) represent the dissipation ability of the electromagnetic wave [27,28]. The electromagnetic loss tangent (tan *δ*_ε_ = *ε*″/*ε*′ and tan *δ*_μ_ = *μ*″/*μ*′) (Figure 6b), the attenuation constant (*α*) (Figure 6c), *C*_0_ (Figure 6d), the typical Cole–Cole semicircles (Figure 6e), and the reflection loss (*RL*) curves (Figure 6f,g) are calculated according to Formulas (1)–(5) [29,30]. The mBPUF samples with a content of 30% are mixed with paraffin and BPU to make a coaxial ring and coating, respectively. The reflection loss (*RL*) depends upon the electromagnetic parameter of the materials, the thickness (*d*), the working frequency (*f*), and the velocity of the electromagnetic wave in a vacuum (*c*). These are constructed according to the transmission line theory. The calculation formulas of *RL* are as follows: (1)$$RL=20lg\frac{\left|Z_{in}-Z_{0}\right|}{\left|Z_{in}+Z_{0}\right|}$$
(2)$$Z_{in}=\sqrt{\frac{\mu_{r}}{\varepsilon_{r}}}tanh\left(j\frac{2\pi }{c}\sqrt{\mu_{r}\varepsilon_{r}}fd\right)Z_{0}$$
where *Z_in_* and *Z*_0_ are the input and free space impedance of the magnetic materials, respectively.

The other important parameters for the absorption performance and its mechanism can be expressed using the following formulas:(3)$$\alpha =\frac{\sqrt{2}\pi f}{c}\times \sqrt{\left(\mu^{\prime \prime }\varepsilon^{\prime \prime }-\mu^{\prime }\varepsilon^{\prime }\right)+\sqrt{{\left(\mu^{\prime \prime }\varepsilon^{\prime \prime }-\mu^{\prime }\varepsilon^{\prime }\right)}^{2}+{\left(\mu^{\prime }\varepsilon^{\prime \prime }+\mu^{\prime \prime }\varepsilon^{\prime }\right)}^{2}}}$$
(4)$$C_{0}=\mu^{\prime \prime }{\left(\mu^{\prime }\right)}^{-2}f^{-1}$$
(5)$${\left(\varepsilon^{\prime }-\frac{\varepsilon_{s}+\varepsilon_{\infty }}{2}\right)}^{2}+{\varepsilon^{\prime \prime }}^{2}={\left(\frac{\varepsilon_{s}-\varepsilon_{\infty }}{2}\right)}^{2}$$

The attenuation constant (α) is the other factor that is typically employed to both estimate and determine the microwave absorption performance (Formula (3)). In addition, *C*_0_ is one of the major magnetic loss originators (Formula (4)), which expresses the eddy current effect [31,32]. The parameters of static permittivity (*ε*_s_) and relative dielectric permittivity (*ε*_∞_) at an infinite frequency (Formula (5)) relate to the permittivity. From Figure 4, the magnetic loss tangent curve is represented by tan *δ*_ε_ and shows no evident change with the modification of the frequency. Within the high-frequency region, magnetic loss plays a major role in the sample. *C*_0_ is a parameter that reflects the eddy current loss in an alternating magnetic field [33]. In Figure 6d, *C*_0_ declines in the lower-frequency range, and then increases slowly to a stable level. 

The results show that the eddy current effect is an important magnetic loss mechanism for the materials. It is produced by electromagnetic conversion, and the pathway is displayed in the dotted line of Figure 6d; when the microwave penetrates into the interior composite, the microwave energy is consumed rapidly by the hierarchical microstructures with dual dielectric–magnetic effects and dissipated into heat energy. The Cole–Cole semicircles are displayed in the form of a *ε*′–*ε*″ curve (Figure 6e) for the mBPUF, which accounts for the existence of the interface polarization loss [34]. A number of interfaces are beneficial for interfacial polarization. Several semicircles in the plot of the *ε*″–*ε*′ curve illustrate the multiple relaxation process with the coexistence of conductivity. 

The *RL* curves in Figure 6f,g of the mBPUF is calculated by the electromagnetic parameters *ε_r_* and *μ_r_* for different thicknesses. The effective *RL* under −10 dB is seen from 4.16 GHz to 18 GHz for a thickness of less than 5 mm, which indicates that the mBPUF powders with low filler loading for the paraffin exhibit excellent microwave absorbency. The surface density of the coating is 0.35 kg/cm^2^, the *RL*_min_ in Figure 6h of the coating is −19.96 dB at 13.15 GHz, and the effective absorption frequency fields (*RL* < −10 dB) are from 11.25 GHz to 15.97 GHz. This is highly consistent with the above result, which tested by the coaxial ring and the base plate film according to a paragraph of test in Appendix A. From the obtained results, it is clear that the mBPUF is an excellent candidate for a lightweight broadband-absorbing material, and the BPU can be used as the novel bio-based matrix in the field of absorbing materials.

### 2.3. Electromagnetic Mechanism 

The microstructures of the component and the complex are constructed by MS.7.0 [35,36,37] and shown in Appendix A, and the microwave absorption mechanism for the mBPUF composite is shown in Figure 7. The good dielectric and magnetic matching effects are the primarily absorption mechanism for the composites [38], namely the combination of dielectric and magnetic losses, which are generated by the multiple reflections, interfacial polarization, magnetic loss, and eddy current effects. First, a porous skeleton for the mBPUF in the composites leads to multiple reflections and attenuates the incident wave [39]. Next, numerous interfaces are produced between the Fe_3_O_4_ and the BPU in the foam skeleton and the membrane skeleton, which enables the transfer of the incident wave into heat via an interfacial polarization [40]. Third, the Fe_3_O_4_ nanoparticles, which have both dielectric and magnetic loss characteristics, are encased in the matrix. This may generate eddy current effects under the alternating electromagnetic field [41]. Furthermore, porous hierarchical microstructures of mBPUF can be regarded as a semiconductor configuration, and plasma can be generated within the pores, which thus dissipates the microwave radiation [42]. Overall, the cooperation of the dielectric and magnetic effects enhances the absorption ability of the composites, and some possible plasma mechanism of microwave absorption is indeed worthy of attention. Herein, the mBPUF composite has a lightweight characteristic and superior microwave-absorbing ability compared with other references, as shown in Figure 8 and Table 1.

## 3. Discussion

In the article, sustainable renewable polymer foam used as a lightweight porous skeleton for microwave absorption is fabricated with structure design. *Sapiumse biferum* kernel oil-based polyurethane foam/Fe_3_O_4_ composites with porous hierarchical structure generating heterogeneous interfaces and abundant porous structures are mainly responsible for enhanced microwave absorption performance. The basic transmission line equations are constructed according to the transmission line theory and a microwave network. All of these effective parameters can be calculated from the corresponding frequency-dependent parameters for bulk components using electromagnetic field theory. It is worth noting that the porous hierarchical microstructures between the magnetic and resin matrix can be regarded as a semiconductor configuration. The plasma will be generated within the pores to dissipate the microwave radiation. This is a new reflection loss mechanism for the porous hierarchical materials.

## 4. Materials and Methods

### 4.1. Materials

The *Sapiumse biferum* kernel oil polyol(SSP, 296 mg KOH·g^−1^) was synthesized with *Sapiumse biferum* kernel oil (SSO, a paragraph of text in Appendix A) in the laboratory (Wuhan, China) using techniques based on previous experience [46]. Stannous octanoate (SnOct, AR, 99.9%), triethylamine (TEA, AR, 99.9%), and polyethyleneglycol400 (PEG-400, AR, 99.5%) were provided by Aladdin Chemistry Ltd Co. (Shanghai, China). Diphenylmethane diisocyanate (pMDI 44v20, TP, NCO% = 30.0–32.0%) and silicone oil were supplied by Chengdu Advanced Polymer Technology Co., Ltd. (Chengdu, China). Dichloromethane (DCM, AR, 99.9%), hexahydrated ferric chloride (FeCl_3_·6H_2_O, AR, 99.7%), ferroussulfate (FeSO_4_·7H_2_O, AR, 99.9%), ammoniumhydroxide (NH_3_·H_2_O, AR, 25.0%~28.0%), and sodium dodecyl benzene sulfonate (SDBS, AR, 99.0%) were purchased from Sinopharm Chemical Reagent Co., Ltd. (Shanghai, China). Deionized water was sourced from the laboratory (Wuhan, China). 

### 4.2. Preparation of Magnetic Fe_3_O_4_ Nanoparticles 

TheFe_3_O_4_ nanoparticles were synthesized according to vacuum coprecipitation [47]. The raw materials, FeSO_4_·7H_2_O and FeCl_3_·6H_2_O, with a molarratio of 1:1.8, were dissolved in 200 mL of deionized water and then stirred with mechanical stirring in water bath, under a nitrogen atmosphere. Next, 100 mL NH_3_·H_2_O solution was added dropwise into the system at 65 °C until the pH reached higher than 12 and continued to react for 1 h. Following this, the system temperature was raised to 85 °C, and the SDBS (1% of the system) was quickly added. The product was then continually stirred until room temperature was achieved. Subsequently, the mixture was separated via a magnetic method and washed with deionized water until neutral. Then, the mixture was freeze-dried to remove the water for 24 h. Finally, the products were ground and then collected for later use. 

### 4.3. Preparation of Magnetic Bio-Based Polyurethane Foam (mBPUF) Composite 

The mBPUF were prepared using SSP and pMDI as the matrix monomers, Fe_3_O_4_ particles with 30 wt% of BPUF as absorbents, deionized water with 4 wt% of SSP as the blowing agent, and PEG-400 with 5% of SSP as the chain extender. The synthesis process of SSP-based BPUF is expressed in Appendix A. The synthesis and molding process for the mBPUF composite is displayed in Figure 1 and Appendix A. The mixture was freely foamed with a flat plate, and then removed and inserted into an oven for vacuum drying at 60 °C for 10 h. Following this, the composite foam was pulverized and removed from the foam powders using 70-mesh sieves.

### 4.4. Characterization 

The structural information of the products was characterized by Fourier transform infrared spectroscopy (FITR, Vertex 70 FTIR) (Bruker Company, Karlsruhe, Germany), which was performed on a spectrometer from 400 cm^−1^ to 4000 cm^−1^ with 4 cm^−1^ resolutions at room temperature. The vertical fracture surface morphology of the mBPUF was evaluated by use of scanning electron microscopy (SEM, Nova Nano SEM 450) and the relative elemental composition information was obtained by employment of an energy-dispersive X-ray (EDSX) spectrometer (FEI Company, Eindhoven, Netherlands). The phase structure of the mBPUF and its composites were acquired from X-ray diffraction (XRD, X’pert3 powder) (PANalytical B.V., Panakot, Netherlands) with a 2*θ* range from 5° to 70° at 17° min^−1^. The distribution of the elements on the surface of the composites was detected by X-ray photoelectron spectroscopy (XPS, AXIS-ULTRA DLD-600) (Shimadzu Kratos, Kyoto, Japan). The surface density was calculated as the ratio of mass to area for a given thickness. The magnetic properties of the products were tested by use of a vibrating sample magnetometer (VSM, Lake Shore 7404) (Lakeshore, Columbus, OH, USA). The electromagnetic properties are tested according to the GJB 2038A-2001 by using vector network analyzer (VNA, PNA-X) (Agilent N5244A, Qingdao, China). Molecular simulation was applied to construct the microstructure of the mixing system by using the Materials Studio v7.0 (MS) (Accelrys Co., Ltd., San Diego, CA, USA).

## 5. Conclusions

This article discussed a convenient and feasible microstructure design ideal to prepare lightweight renewable bio-based mBPUF composites with potential broadband microwave-absorbing performance. The microstructures and the performance of the prepared samples were systematically characterized. It was found that the porous mBPUF embedded with Fe_3_O_4_ nanoparticles exhibited heterostructures and could be used as a functional filler in the BPU matrix. Due to the heterostructure and porous microstructures of the lightweight mBPUF, the multiple reflections, interfacial polarization, magnetic loss, eddy current effects, and plasma were produced when subjected to magnetic fields. The mBPUF composite with 9% content of Fe_3_O_4_ exhibited outstanding microwave absorbency with an effective bandwidth of 4.62 and 4.72 GHz at a thickness of 1.789 mm and 2.0 mm in paraffin and BPU matrix, respectively, which indicated good stability in different matrixes. The effective absorbing frequency range reached 13.84 GHz when the thickness was less than 5 mm. The good impedance matching effect makes the mBPUF a promising lightweight broadband-absorbing material, which will expand the practical use of renewable resources and achieve great economic value. 

## Data Availability

The data presented are contained within the article or Appendix A. The data are not publicly available due to privacy.

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
