# Peer review of "Synthesis of Lightweight Renewable Microwave-Absorbing Bio-Polyurethane/Fe3O4 Composite Foam: Structure Analysis and Absorption Mechanism"

_ijms, 2022, doi:10.3390/ijms232012301_

Round 1

Reviewer 1 Report

The manuscript “Lightweight renewable microwave absorbing foam from Sapium sebiferum kernel oil-based polyurethane: hierarchical structural analysis and mechanism elucidation” by X. Xu et al. is devoted to the synthesis and characterization of a microwave-absorbing composite material with a foam structure using an oil-based polyurethane and ferrite nanoparticles. The obtained results may be of interest to specialists in the field of design and synthesis of antireflection coatings for the microwave range; the studies were carried out at an appropriately high scientific and technical level. However, when reading the manuscript, some questions arise regarding the interpretation of the expected and observed effects and the presentation of the results obtained. In my opinion, these questions should be answered before a decision is made to accept the manuscript.

1. The authors pay considerable attention to various methods for analyzing the synthesized composite at the atomic and molecular levels (FTIR, XRD, SEM, EPD), while desirable data on the electrical and magnetic properties of the material at the macroscopic level are presented in much smaller quantities. Actually, only the magnetization curves for the samples are displayed in Fig. 5 (a). It follows from the text of the manuscript that the rest of the graphs in Figure 5 are theoretical curves obtained by calculation. How were the effective values of the frequency-dependent permittivity and permeability set in the calculations? For such heterogeneous systems in the microwave range, these effective parameters can be calculated from the corresponding frequency-dependent parameters for bulk components using an acceptable effective medium model. What effective medium model did the authors use? These and other related issues should be discussed in much more detail, with appropriate references included in the bibliography.

2. The subsection 2.3 (“Electromagnetic mechanism”) raises many questions. First, the authors claim multiple reflections of an electromagnetic wave inside the layer from its boundaries (Fig. 6, left part). It is not clear why there are no reflected and transmitted components in the figure after double, triple, etc. reflections inside a coating? Such multiple reflections, it seems, should lead to a comb structure of the dependences in Fig. 5, f at a fixed layer thickness. In addition, the claimed eddy current effect in such structures should be discussed in much more detail and using quantitative estimates, taking into account the filling of the volume of the structure with ferrite particles and, accordingly, the efficiency of electromagnetic couplings between them. In general, this section requires a more convincing justification with quantitative estimates.

3. The paper does not present any experimental data confirming the results of calculation of the electromagnetic properties (Subsection 3.4).

Finally, the manuscript should be substantially revised before the acceptance. 

Author Response

Revision for Reviewer 1

    Thank you for your much attention and hard working on this manuscript. Many thanks for pointing out the weakness of the manuscripts. The following revision explanations are account for the opinions. The detailed reply is provided in the attachment word. 

Reviewer 2 Report

 Review of the manuscript by Referee -1.

Lightweight renewable microwave absorbing foam from Sapium sebiferum kernel oil-based polyurethane: hierarchical structural analysis and mechanism elucidation

by

Xiaoling Xu , Xiaoke Tian , Guangxu Bo , Xingjian Su , Jinyong Yan, Yunjun Yan 

I read the paper with interest.  I believe it should be published subject to a revision.  My comments and the required alterations, corrections, amendments and clarifications are provided below.  I am happy to re-review the paper after the revision.

1.     Although the title is relatively long, it should include the main component of microwave absorbing composite, namely Fe3O4.  I suggest a title such as:

 Synthesis of lightweight renewable microwave absorbing bio-polyurethane /  Fe3O4  composite foam:   structural analysis and mechanism of reflection loss

2.     Abstract:

Fe3O4  is referred to as ferrite and in the text it is changed to ferro oxide (Section 3.2).  Strictly speaking both terms are inaccurate to describe pure Fe3O4 unless Fe3O4  is not pure and contains other oxides.  Fe3O4 should be referred to as magnetite or simply Fe3O4 (spinel).  

3.     Introduction

(a)    Some of the acronyms are used without definition.  For example TPU/G  or TPU/G-3.  Some of them are defined later on such as TPU (in page 2) while others such as BPUF are defined a number of times.   Acronyms should be defined in the Introduction-Section as they first appear.

(b)    Introduction does not contain some of the important relevant references indicating that the literature survey is incomplete and there is a lack of recent relevant developments.  The references should include recent reviews on the subject such as the following:

 (Ref-1) B. Wang, Q. Wu, Y. Fu, T. Liu, A review on carbon/magnetic metal composites for microwave absorption, J. Mater. Sci. Technol., 86 2021, 91-109.  (This review is an update of the above publication focussing on carbon/graphene supports and metals, including Fe).

 (Ref-2) Zheng, W.; Ye, W.; Yang, P.; Wang, D.; Xiong, Y.; Liu, Z.; Qi, J.; Zhang, Y. Recent Progress in Iron-Based Microwave Absorbing Composites: A Review and Prospective. Molecules 2022, 27, 4117. https://doi.org/10.3390/ molecules27134117.  (This is an excellent and very recent review and highly relevant to this manuscript)

 (c)     Based on microwave induced plasma generation, a new mechanism of microwave absorption has recently been discovered.  See Reference 31 in the publication (Ref-2) and detailed below.

(Ref-3) Akay, G. Plasma Generating—Chemical Looping Catalyst Synthesis by Microwave Plasma Shock for Nitrogen Fixation from Air and Hydrogen Production from Water for Agriculture and Energy Technologies in Global Warming Prevention. Catalysts, 10, 2020, 152.

(d)    According to the above reference (Ref-3) when metal oxides with the pores of an insulator (thus forming a semiconductor configuration) plasma is generated within the pores which thus dissipates the microwave radiation.  This method of reflection loss through plasma generation represents a novel and the mechanism of microwave absorption  in the current manuscript should also consider this possibility. 

 (e)    Figure 1 is not an absorption spectrum but a transmission spectrum.

 (f)      In Figure 1, there are several other identifiable peaks for Fe3O4.  For example, peak at 1629 cm-1 is assigned to  Fe-O.  See Ref-4 below:

(Ref-4) Lu, W. J. Magnetism  Magnetic Materials, 32(2010) 1828-1833.

 (g)    Figure 2: XRD pattern of Fe3O4 indicates that the crystallize size based on the dominant reflection (311) must be very large, I guess ca. 30-50 nm. The size should be evaluated using Debye-Scherer equation.

 (h)    XRD pattern of the composite mBPUF shows reduced crystallize size as well as shift in diffraction angle.  These should be explained and crystallize size based on the dominant peak associated with reflection (331) should be evaluated.

 (i)      BPUF is, like other polyurethanes, can be expected to be amorphous.  This could have been confirmed experimentally.  However, Figure 2 shows a peak at ca. 230 which should be associated with the base polymer.  These results indicates that macromolecules form a shell (bound polymer) around Fe3O4 particles and due to the chain conformational restrictions, they show crystalline order.  These results can be explained by morphology of such bound  polymer (ca. 2-3 nm thick)  as studied in (Ref-5):

(Ref-5)  Akay, G. Polymer Engineering and Science.  30 (1990) 1361-1372.

(j)      I therefore disagree with the statement on Page 3 (Last sentence of Para.1) that: “It indicates that part of the magnetic absorbents is embedded in the porous resin which hides the crystalline structure.”  

(k)      Infact Fig. 3 shows that the overwhelming part of the particles are in the pores not within the walls of the pores.

 (l)  Fig. 5 (a) The y-axis  should read “Magnetic Moment (emu/g)” rather than “Monent”.

 (m)    Section 2.3. Electromagnetic mechanism

 This section should include some discussion as regards the plasma generation by microwave absorption in stealth mechanism as indicated in Para. (c).

Author Response

    Thank you for your much attention and hard working on this manuscript. Many thanks for pointing out the weakness of the manuscripts. The following revision explanations are account for the opinions. The detailed reply is provided in the attachment word.

Reviewer 3 Report

Dear Authors, congratulations on the well-written and clear manuscript. The characterizations are detailed. Here are some considerations for you to improve the manuscript.

In the first sentence of the introduction, Other than Polaris and lightweight materials it also needs to be laced with absorbing materials.

In the second paragraph consider changing “a plenty of” to “many”. Change “some” to “there are”. Remove “has been scored”. 

In the introduction, you mentioned the data that you have shown later in table 2. You can shorten your introduction to make it more succinct.

Do quote the last second and third sentences of the first paragraph in 2.1. Instead of using the word “transfer” consider using “shifts”. Instead of using “relates” use “correlate”.

In Figure 1, use the same small font size and label the peaks using arrows.

Take note of the spacing of the lines on page 3 and check if it is intentional. Instead of “most” you can consider writing “major” XRD peaks are present.

In Figure 3, since EDS was performed, it would be interesting to show the elemental distribution in the image.

In Figure 4, it would be good to half a legend to what are the colour codes.

Take note that the y-axis of Figure 4B is not legible.

Do quote the calculation formulas.

In figure 5, the moment is spelt wrongly.

In 3.2, why was the ratio 1:1:8 chosen for synthesis?

Include the keywords such as sustainable renewable lightweight porous in your conclusion. In addition, reiterate the advantage of your material compared to other manuscripts.

Author Response

(The authors gave the same response as above.)

Round 2

Reviewer 1 Report

Introduced revisions improved the manuscript quality and I can recommend to accept it.